# Safety and Immunogenicity of OVX836, a Nucleoprotein-Based Universal Influenza Vaccine, Co-Administered with Fluarix^®^ Tetra, a Seasonal Hemagglutinin-Based Vaccine

**DOI:** 10.3390/vaccines13060558

**Published:** 2025-05-23

**Authors:** Nicola Groth, Jacques Bruhwyler, Jessika Tourneur, Emilie Piat, Philippe Moris, Alexandre Le Vert, Florence Nicolas

**Affiliations:** Biometry and Medical Writing, Osivax, 70 Rue Saint-Jean-de-Dieu, 69007 Lyon, France; ngroth@osivax.com (N.G.); jtourneur@osivax.com (J.T.); epiat@osivax.com (E.P.); pmoris@osivax.com (P.M.); alevert@osivax.com (A.L.V.); fnicolas@osivax.com (F.N.)

**Keywords:** influenza, universal vaccine, Phase 2a, healthy participants, hemagglutinin, seasonal vaccine, nucleoprotein, safety, immunogenicity, vaccine combination

## Abstract

**Background/Objectives**: The combination of a hemagglutinin antigen (HA)-based inactivated influenza vaccine (IIV; Fluarix^®^ Tetra; GlaxoSmithKline) with a nucleoprotein (NP)-based vaccine, such as OVX836, should increase the efficacy of influenza vaccines since it leverages two complementary immunological mechanisms: HA antibodies targeting the virus envelope and neutralizing it, and an NP cell-mediated immune (CMI) response destroying infected cells. **Methods**: This was a randomized, double-blind, Phase 2a study (ClinicalTrials.gov NCT05284799) including three groups of 60 healthy subjects (18–55 years old) receiving either IIV + placebo, IIV + OVX836 (480 µg), or OVX836 + placebo intramuscularly and concomitantly into the same deltoid muscle. The endpoints were reactogenicity, safety, and immunogenicity (hemagglutination inhibition assay [HAI], anti-NP immunoglobulin G [IgG], and NP-specific cell-mediated immunity [CMI]). **Results**: The co-administration of IIV + OVX836 was safe and well-tolerated. The HAI response was strong and similar in the two IIV groups with no interference of OVX836. The humoral anti-NP IgG and NP-specific CMI responses to OVX836 were strong in the two OVX836 groups, and no major interference of IIV was observed. **Conclusions**: This study supports further clinical development of OVX836 as a combined IIV/OVX836 seasonal vaccine capable of inducing robust and complementary HAI and CMI NP-specific responses.

## 1. Introduction

Vaccination remains the most effective preventive measure against influenza virus transmission and associated morbidity and mortality [1]. Current influenza vaccines, which induce antibody responses against virus surface glycoproteins (hemagglutinin [HA] and neuraminidase), exhibit sub-optimal effectiveness. Not only does the effectiveness of influenza vaccines vary from year to year, but for young, adult, and elderly populations, it is consistently below the National Institute on Allergy and Infectious Diseases and World Health Organization target of 75% [2,3,4]. Furthermore, computational simulation modeling of the spread of COVID-19 predicted that at least 70% vaccine efficacy is needed to prevent an epidemic [5]. The extrapolation to other respiratory viruses is not implicit, but the current efficacy/effectiveness of seasonal influenza vaccines is nevertheless well below 70%.

While the antibody response to influenza vaccines has been the main focus of protection, a cell-mediated immune (CMI) response, in particular CD4+ and CD8+ T-cells, towards the well-conserved internal influenza nucleoprotein (NP) could be an alternative and complementary path for protection. This approach is supported by epidemiological evidence correlating the NP CMI responses to a lower risk of influenza disease [2,6,7,8,9,10].

The OVX836 vaccine candidate (Osivax, Lyon, France) is a recombinant protein that includes the full-length nucleoprotein (NP) of the A/WSN/1933(H1N1) influenza virus, combined with OVX313 (oligoDOM^®^), OSIVAX’s proprietary self-assembling nanoparticle technology. This recombinant protein spontaneously assembles into positively charged nanoparticles, each consisting of a circular disposition of seven copies of the NP fused to OVX313 [11,12].

The safety, humoral response, and CMI of OVX836 have been assessed in two Phase 2a [13,14] studies, both in young and elderly adults. Across these studies, OVX836 demonstrated a favorable safety profile up to 480 µg, with no dose-dependent increase in reactogenicity and no indication of reaching the maximum tolerated dose [13,14].

In Phase 2a studies, significant NP-specific humoral (anti-NP IgG) and cellular (interferon gamma [IFNγ]-secreting cells) immune responses were observed at three dose levels (180 µg, 300 µg, and 480 µg), without a clear dose–response relationship. A CD8^+^ T-cell response was detected only at the 300 µg and 480 µg dose levels in 18–55-year-old subjects [13]. T-cell responses were shown to be highly cross-reactive against various influenza A strains, including seasonal and highly pathogenic avian strains [15], an observation related to the fact that NP is highly conserved across all A influenza strains [15]. In addition, a signal of vaccine efficacy (VE) of 78% (95% confidence interval: 3.6–95.3%) was observed in an epidemiological context of H3N2 circulation [14].

The simultaneous induction of neutralizing antibodies, antibody effector functions, and T-cell immunity may constitute the most effective vaccination strategy, both in pandemic and seasonal influenza situations [16]. In this context, the association of an HA-based inactivated influenza vaccine (IIV) with an NP-based vaccine, such as OVX836, could increase the efficacy of influenza vaccines through two different and complementary immunological mechanisms: HA antibodies targeting the virus envelope and neutralizing the virus and an NP-specific CMI response destroying infected cells to prevent further viral spreading [2]. In a preclinical model, OVX836 protected mice against viral challenge with three different influenza A subtypes, isolated several decades apart, and reduced viral load. When OVX836 and IIV were co-administered, the combination was even more effective in reducing lung viral load [12], which is the first preclinical proof of concept for the proposed approach.

Prior to initiating large-scale field efficacy trials and investing in the development of combination formulations, it is essential to demonstrate safety and the absence of immune interference between IIV and OVX836.

The aim of the present Phase 2a study was to assess the safety and immunogenicity (humoral and cellular) of OVX836 and IIV (Fluarix^®^ Tetra, a GlaxoSmithKline commercial seasonal quadrivalent influenza vaccine), administered concomitantly into the same deltoid muscle (to ensure drainage of the two vaccines by the same lymph nodes), in adults aged 18 to 55 years.

## 2. Methods

This randomized, double-blind, parallel-group Phase 2a study (OVX836-004; ClinicalTrials.gov NCT05284799) was performed in a single center (Northern Beaches Clinical Research [NBCR], Suite 201, Level 2, 694-696 Pittwater Road, Brookvale 2100, Australia) in accordance with Good Clinical Practice. It was approved by the Australian Human Research Ethics Committees, followed by the Clinical Trial Notification acknowledgment of the Australian Therapeutic Goods Administration. All participants gave their written informed consent.

Three groups of 60 subjects each were randomized (1:1:1) to receive either Fluarix^®^ Tetra (GlaxoSmithKline, London, UK) and a placebo, Fluarix^®^ Tetra and OVX836 (480 µg), or OVX836 (480 µg) and a placebo, intramuscularly (IM) and concomitantly into the same deltoid muscle (2.5 cm apart) of the non-dominant arm. OVX836 included the full-length nucleoprotein (NP) of the A/WSN/1933(H1N1) influenza virus. Fluarix^®^ Tetra (abbreviated IIV hereafter) was an inactivated and purified split influenza vaccine. The antigen composition and strains for the 2022 influenza season were 15 µg of A/Victoria/2570/2019 (H1N1)pdm09-like virus, 15 µg of A/Darwin/9/2021 (H3N2)-like virus, 15 µg of B/Austria/1359417/2021-like (B/Victoria lineage) virus, and 15 µg of B/Phuket/3073/2013-like (B/Yamagata lineage) virus. The placebo consisted of a saline solution (NaCl 0.9%).

A total of 180 healthy (as determined by medical history and medical examination) male and female subjects, aged 18–55 years, were identified from the NBCR’s database and included in the study. Subjects were fully vaccinated with a licensed SARS-CoV-2 (COVID-19) vaccine according to the national recommendations. In females of childbearing potential, a urine sample was collected on Day 1 to check the absence of pregnancy. The inclusion and exclusion criteria, as listed in the clinical study protocol, can be found in Appendix A.

After the vaccination (Day 1), the subjects were kept under observation on site for 30 min before being discharged. This was followed by a 7-day period wherein solicited local (injection site pain, redness, and swelling) and systemic (fatigue, headache, arthralgia, malaise, myalgia, and fever) signs and symptoms were collected using an eDiary, a 29-day period for reporting unsolicited adverse events (AEs), and a 180-day period (end of study) for reporting serious adverse events (SAEs). The severity of AEs was assessed in accordance with the Food and Drug Administration’s (FDA) guidance (2005) on the Toxicity Grading Scale for Healthy Adult and Adolescent Volunteers Enrolled in Preventive Vaccine Clinical Trials. All subjects visited the investigator’s site on Day 1 and Day 29, and a phone contact between the investigator and the subject was set up on Day 8 and Day 180 post-injection. Approximately 50% of the subjects were allocated to a subset to undergo peripheral blood mononuclear cell (PBMC) sampling. These subjects also visited the site on Day 8.

Blood samples (10 mL of whole blood per visit) to collect serum for HA and NP humoral immunity assessment were drawn on Day 1 (all subjects), Day 8 (only in the PBMC subset), and Day 29 (all subjects). In the PBMC subset, an additional sample of 50 mL of whole blood was drawn to collect PBMCs for the evaluation of NP-specific CMI on Day 1 and Day 8. The immunoassays are described in Appendix A. An electronic Case Report Form (eCRF) was used for data collection.

The first primary endpoint was the HA immune response in the two groups receiving IIV, as measured by the hemagglutination inhibition assay (HAI), for each of the four influenza strains contained in the IIV. The following endpoints were considered: (i) seroconversion rate at Day 29 versus pre-injection baseline (Day 1), seroconversion being defined as a negative pre-vaccination HAI titer and post-vaccination HAI titer ≥1:40, or a fourfold increase in HAI titer between pre- and post-vaccination timepoints; (ii) seroprotection rate, defined as the proportion of subjects achieving a titer ≥1:40 at Day 29; and (iii) HAI titers Day 29/Day 1 geometric mean ratios (GMRs). To reach the primary endpoint (Note for Guidance on Harmonization of Requirements for Influenza Vaccines, CPMP/BWP/214/96, 1997), at least one of the three following criteria (not necessarily the same for each strain) had to be achieved on Day 29 for the four influenza strains: (i) seroconversion rate of at least 40%, (ii) ≥70% of subjects achieving a titer ≥1:40, and/or (iii) GMR > 2.5. According to the FDA’s Guidance for Industry, Clinical Data Needed to Support the Licensure of Seasonal Inactivated Influenza Vaccines (2007), the lower limit (LL) of the 95% confidence interval (95% CI) has to be above 40% for the seroconversion rate and at least 70% for the seroprotection rate. These readouts were also considered.

The second primary endpoint was the safety evaluation in the three treatment groups, i.e., (i) number and percentage of subjects reporting solicited local and systemic signs and symptoms during 7 days after vaccine administration; (ii) number and percentage of subjects reporting unsolicited AEs during 29 days after vaccine administration; and (iii) number and percentage of subjects reporting SAEs during the whole study duration.

The secondary endpoints were: (i) the HAI geometric mean titers (GMTs) on Day 1 (pre-injection baseline) and Day 29 for each of the four strains contained in the IIV; (ii) the change in NP-specific T-cell frequencies in PBMCs, measured by IFNγ ELISPOT, at Day 8 versus pre-injection baseline (Day 1); (iii) the GMTs of anti-NP IgG; and (iv) the number and percentage of subjects with a four-fold increase in anti-NP IgG titer on Day 29 with respect to pre-injection baseline (Day 1).

The exploratory immunogenicity endpoints were the NP-specific CD4+ and CD8+ T-cell percentages measured by flow cytometry, identified as expressing markers such as IFNγ, interleukin-2 (IL-2), and/or tumor necrosis factor alpha (TNFα), upon in vitro stimulation with an NP peptide pool at Day 1 (pre-injection baseline) and Day 8.

The estimation of the sample size was based on the CPMP/BWP/214/96 (1997) for the demonstration that at least one of the following criteria of acceptance of seasonal influenza vaccines was achieved in the groups receiving the IIV. The above-mentioned guidelines recommended having at least 50 evaluable subjects per group. In conclusion, the total sample size of evaluable subjects at Day 29 had to be equal to 100 (group IIV/placebo + group IIV/OVX836). Accounting for potential dropouts, 120 subjects were vaccinated in these two groups. The third group (OVX836/placebo) also included 60 subjects.

The primary cohort for the analysis of safety included all subjects that received the vaccines and/or placebo (safety cohort [SC]). The per protocol (PP) analysis included all subjects without significant protocol deviations that received the vaccines and/or placebo and had eligible baseline and complete post-administration blood samples for immunogenicity analysis up to Day 29. The statistical methodology is described in Appendix A.

## 3. Results

### 3.1. Study Population Demographics and Baseline Characteristics

Vaccinations were performed in May/June 2022, during the Southern Hemisphere influenza season, which occurred from March to October 2022 (according to Australian health authorities). The last participant’s phone contact (Day 180) took place in December 2022. A total of 180 participants were enrolled and received the study vaccines and/or placebo, with 60 participants assigned to each treatment group. The PP cohort consisted of 143 participants (48 in the IIV + placebo group, 49 in the IIV + OVX836 group, and 46 in the OVX836 + placebo group) (Figure 1). A total of 151 subjects completed the entire study (up to Day 180): 49 in the IIV + placebo group, 51 in the IIV + OVX836 group, and 51 in the OVX836 + placebo group.

Participants were 37.8 ± 12.8 years old (mean ± SD) (range: 18–55 years). The majority of subjects were females (59.4%) and White-Caucasians (81.1%). The baseline characteristics were comparable across the three treatment groups (Table 1).

### 3.2. Reactogenicity and Safety

Pain, essentially mild and of short duration (2–3 days), was the most frequently reported local symptom, affecting 51.7%, 86.7%, and 80.0% of subjects in the IIV + placebo, IIV + OVX836, and OVX836 + placebo groups, respectively (*p* > 0.05). No subjects reported severe (grade 3) local solicited signs or symptoms.

Myalgia (45% to 65% of subjects) and fatigue (45% to 57% of subjects) were the most frequently reported systemic symptoms. Severe fatigue and severe myalgia were each reported in one subject (1.7%) in the IIV + placebo group. Severe headache was reported by one subject (1.7%) in the OVX836 + placebo group.

A total of 36.7%, 31.7%, and 33.3% of subjects reported at least one occurrence of unsolicited AE (considered related or not to the vaccines/placebo) in the IIV + placebo, IIV + OVX836, and OVX836 + placebo groups, respectively (*p* > 0.05). All these AEs were mild to moderate. No severe or life-threatening AEs were reported.

Three SAEs were reported in three subjects (1.7%) during the safety follow-up (Day 29 and Day 180): basal ganglia infarction in one subject of the IIV + placebo group, appendicitis in one subject of the IIV + OVX836 group, and joint injury in one subject of the IIV + OVX836 group. None of the SAEs was considered related to the vaccines or placebo (Table 2).

### 3.3. Hemagglutination Inhibition Assay

OVX836 administered with placebo did not induce any HAI response (except for the B/Phuket/3073/2013-like [B/Yamagata lineage]), whereas significant HAI responses were observed for the four strains in the groups receiving IIV (Table 3, Figure 2, and Appendix A). The primary immunogenicity objective of the study was met: at least one of the three requested HAI CPMP criteria was achieved on Day 29 for the four influenza strains in the two groups receiving the IIV. In addition, concerning the FDA criteria, it should be emphasized that the LLs of the 95% CI around midpoint seroconversion rates were all above the limit of 40%, except for the A/Darwin/9/2021 (H3N2) strain (midpoint below the threshold of 40% for both groups). The LL of the 95% CI around midpoint seroprotection rates were all above the threshold of 70%, with the exception of the B/Austria/1359417/2021-like (B/Victoria lineage) strain in both IIV groups and the A/Darwin/9/2021 (H3N2) (LL of the 95% CI at 67%) and B/Phuket/3073/2013-like (B/Yamagata lineage) (LL of the 95% CI at 58%) strains in the IIV + placebo group (Table 3). It should also be emphasized that the HAI results achieved in the IIV + placebo and IIV + OVX836 groups were similar. A statistically significant difference in favor of the IIV + OVX836 480 µg group compared to the IIV + placebo group was found for the strains A/Darwin/9/2021 (H3N2) (*p* = 0.047) and A/Victoria/2570/2019 (H1N1)pdm09 (*p* = 0.024) but not for the B strains (*p* > 0.05) (Figure 2).

### 3.4. Cell-Mediated Anti-NP Immune Response

No NP-specific responses were detected in the IIV + placebo, whereas OVX836 induced a strong NP CMI response (*p* < 0.0001), as measured through IFNγ ELISPOT, whether or not administered concomitantly with IIV (Figure 3 and Appendix A). The difference between Day 1 and Day 8 in the number of NP-specific IFNγ spot-forming cells (SFCs) per million PBMCs was 164 ± 190 in the IIV + OVX836 group, versus 178 ± 153 in the OVX836 + placebo group. Between Day 1 and Day 8, the number of NP-specific IFNγ SFCs per million PBMCs increased 6.5-fold (±9.2) and 9.1-fold (±10.4) in the IIV + OVX836 group and OVX836 + placebo group, respectively. No statistically significant difference was found between the two groups (*p* > 0.05).

As measured by intracellular cytokine staining, statistically significant (*p* = 0.0001) increases in the frequencies of NP-specific CD4+ T-cells (% per total CD4+ T-cells) identified as expressing at least IFNγ or at least one cytokine (among IFNγ, IL-2, and TNFα), were observed between Day 1 and Day 8 in the IIV + OVX836 and OVX836 + placebo groups (Figure 4 and Appendix A). There was no significant difference between the two groups for these two readouts (*p* > 0.05). In addition, the polyfunctionality of the cells (in particular, IFNγ+/IL-2+ [*p* = 0.0001] and polypositive [expressing at least two cytokines; *p* < 0.05] CD4+ T-cells) was observed for both groups.

There was no statistically significant increase in the percentages of NP-specific CD8+ T-cells in the three treatment groups of this study (*p* > 0.05; Appendix A).

### 3.5. Humoral Anti-NP Immune Response

No NP-specific humoral response was measured in the IIV + placebo group. OVX836 induced a strong humoral immune response (*p* < 0.0001), whether or not administered concomitantly with IIV. On Day 29, the anti-NP IgG GMT was significantly (*p* = 0.0003) higher in the OVX836 + placebo group (GMT 22,019) compared to the IIV + OVX836 group (GMT 13,738) (Figure 5 and Appendix A). The percentage of subjects presenting a 4-fold increase in the anti-NP IgG titers was significantly (*p* = 0.047) higher in the OVX836 + placebo group (84.8%) than in the IIV + OVX836 group (67.3%).

## 4. Discussion

OVX836 (480 µg) appeared safe and well-tolerated when concomitantly administered intramuscularly in the same arm with an IIV (or with a placebo), thereby corroborating previously published results on OVX836 alone [13,14]. This is an essential prerequisite for pursuing the development of the new proposed prophylactic influenza vaccine, combining an IIV (either trivalent [TIV] or quadrivalent [QIV]) with OVX836. Indeed, the increased reactogenicity associated with the combination of vaccines has always been a major concern for both vaccine manufacturers and health authorities [17,18,19].

The IIV co-administered with OVX836 triggered a strong HAI response, similar to the IIV co-administered with placebo, allowing us to conclude to the absence of interference of OVX836 on the IIV humoral immune response. This is an essential condition for a combined injection of an IIV with OVX836. The primary immunogenicity objective of the study was met, at least one of the three requested CPMP HAI criteria being achieved on Day 29 for the four influenza strains in the two groups receiving the IIV.

In the OVX836 groups, there was a strong and statistically significant CMI response to NP at Day 8 versus Day 1 as measured through IFNγ ELISPOT or intracellular staining, in terms of number of NP-specific IFNγ SFCs per million PBMCs and percentages of NP-specific CD4+ T-cells. No major differences were found between the IIV + OVX836 and OVX836 + placebo groups for the number of NP-specific IFNγ SFCs per million PBMCs and percentages of NP-specific CD4+ T-cells, allowing to conclude to the absence of significant interference of the IIV on the NP-specific CMI response to OVX836.

Contrary to the observation made in a previous study [13], no CD8+ T-cell response could be measured in the vaccine groups receiving OVX836 in the current study. The main reason for this discrepancy could be related to several differences between the two studies: (i) Vaccinations were performed before the influenza season in the Leroux-Roels et al. [13] study, but during the influenza season in the present study. High heterogeneity in baseline CD8+ T-cell percentages was found (coefficient of variation of 140% versus 90% in the previous study [13]), which might be related to the circulation of natural influenza viruses during vaccination and might result in greater difficulty in detecting a CD8+ T-cell response to the vaccine. (ii) The study populations were enrolled in different hemispheres with potential differences in historical influenza virus exposure: the Northern Hemisphere for the Belgian population in the Leroux-Roels et al. [13] study versus the Southern Hemisphere for the Australian population in the present study. (iii) Last but not least, there were more females (70.8%) enrolled in the Leroux-Roels et al. [14] study than in the current study (59.4%). In the paper of Jacobs et al. [14], we have shown that females tended to reach higher humoral and CMI responses to OVX836 than males.

A strong and statistically significant anti-NP IgG response at Day 29 versus Day 1 was measured in the groups receiving OVX836 concomitantly administered with an IIV or placebo. There was a limited interaction (but statistically significant) of the IIV on the anti-NP IgG response to OVX836 on Day 29, the humoral immune response being higher in the OVX836 + placebo group compared to the IIV + OVX836 group.

The results of our study are consistent with those published by Antrobus et al. [20], where co-administration of a seasonal IIV (TIV) and a modified vaccinia virus Ankara-NP + matrix protein M1 (MVA-NP+M1) elicited potent HA humoral and NP+M1 CMI responses. They found that the co-administration group had higher GMRs against the H3N2 vaccine strain than the IIV + placebo group. For the H1N1 and B strains, the differences were smaller and did not reach statistical significance. Consistent with our results, seroconversion and seroprotection rates were similar between the two groups. Additionally, there was no detectable NP+M1 T-cell response (IFNγ ELISPOT) in the IIV + placebo group. In contrast, and also aligned to our results, the MVA-NP+M1 and MVA-NP+M1 + IIV groups induced a strong NP+M1 response, with no statistically significant differences between these two groups.

Although a significant difference in the anti-NP IgG response was found 28 days post-vaccination in the IIV + OVX836 group compared to the OVX836 + placebo group, the amplitude of the humoral response remained high compared to the IIV + placebo group. The relevance of this finding is unknown, as anti-NP antibodies are not considered essential for protection because they do not possess neutralizing activity. However, they could bind to virus-infected cells and mediate protection through Fc-dependent functions [10,21,22,23].

The absence of significant interference of OVX836 on the HA humoral response to the IIV, as well as the limited interference of the IIV on the NP CMI response to OVX836, support the second prerequisite for a combination anti-HA and anti-NP influenza vaccine.

These results have since been extended in a larger (N = 477 subjects) randomized, placebo-controlled, Phase 2a trial (ClinicalTrials.gov NCT05734040) conducted during the 2023 influenza season in Australia. The study investigated the response to two IIVs (Fluarix^®^ Tetra and Afluria Quad^®^) when administered alone and concomitantly with OVX836 in contralateral deltoid muscles of the upper arms. The manuscript of this study is also being prepared for publication.

## 5. Conclusions

The results (safety and immunogenicity) from this study and previous studies [13,14,24] support the development of OVX836 as a stand-alone universal influenza vaccine and as a combined IIV/OVX836 vaccine capable of inducing robust and complementary antibody (HAI) and CMI NP responses. This could potentially lead to higher efficacy due to synergistic immune responses and broader cross-protection to better prepare for the years of influenza strain mismatch and eventually for the shift to new pandemic strains.

Concomitant administration still requires two separate injections and might therefore result in implementation challenges and raise vaccine hesitancy concerns related, among others, to the fear of needles [25,26,27] widely represented in the population. This could, in turn, lower the vaccine coverage rate of the co-administration of the two vaccines and hence the expected public health benefit of combining the two products. As such, Osivax is pursuing the clinical development of the combination with an extemporaneous mix of OVX836 with a standard dose of IIV into a single injection.

## Figures and Tables

**Figure 1 vaccines-13-00558-f001:**
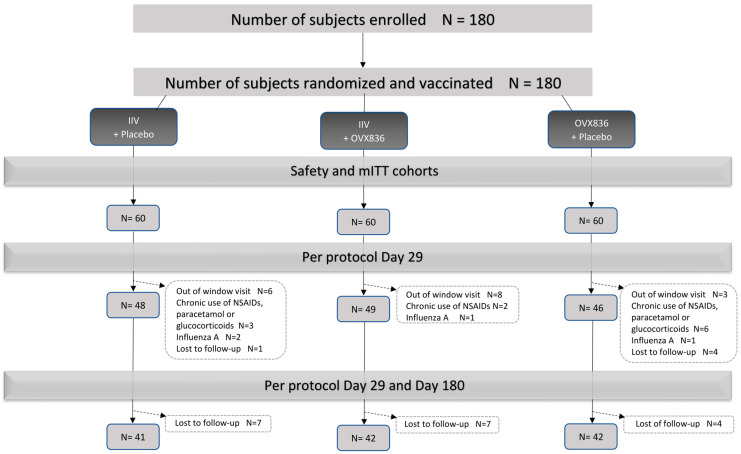
CONSORT diagram of the OVX836-004 study.

**Figure 2 vaccines-13-00558-f002:**
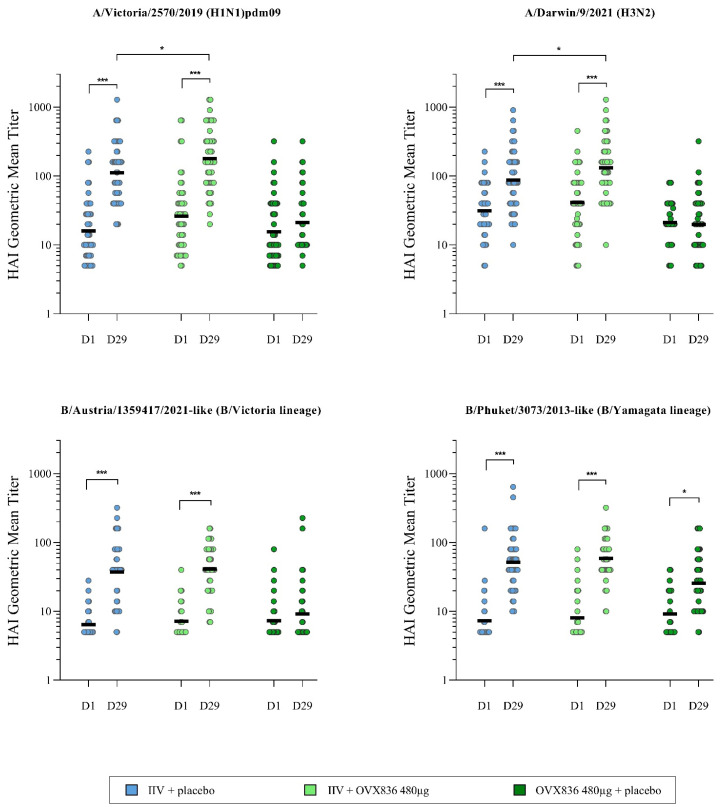
Hemagglutination inhibition geometric mean titers (GMTs) at pre-injection baseline (Day 1) and Day 29 for the four influenza strains in the three treatment groups (per protocol cohort). Results are shown as individual data and GMTs (horizontal bar). Intragroup comparisons (paired Student’s *t*-tests; * *p* < 0.05; *** *p* < 0.001) and intergroup comparisons (ANOVA followed by Bonferroni’s tests; * *p* < 0.05).

**Figure 3 vaccines-13-00558-f003:**
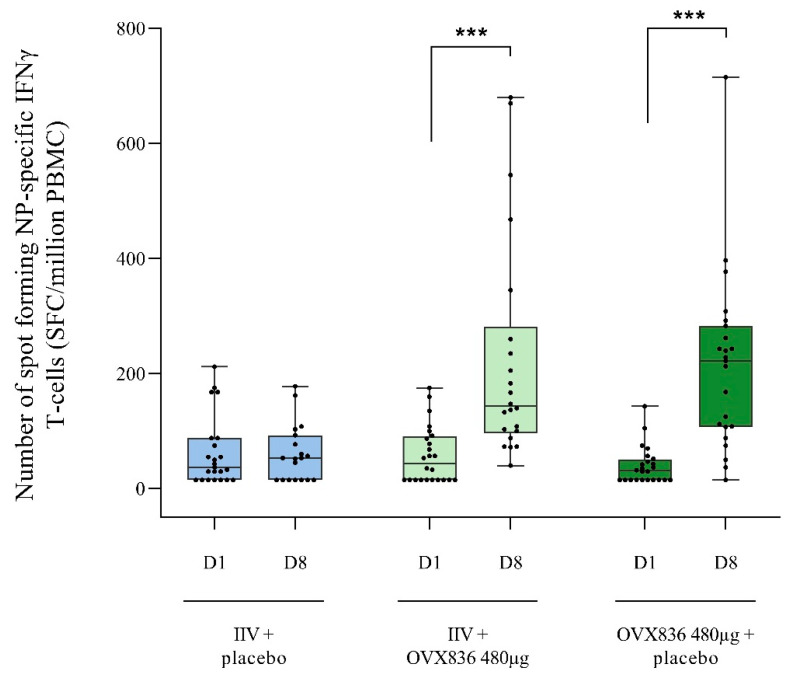
Box plot of the number of NP-specific IFNγ spot-forming cells (SFC/million PBMC) at Day 1 (pre-vaccination) and Day 8 in the three treatment groups (per protocol cohort). Results are shown as individual data, median (horizontal bar), interquartile interval (box), and minimum and maximum values. Intragroup comparisons (paired Student’s *t* tests; *** *p* < 0.0001).

**Figure 4 vaccines-13-00558-f004:**
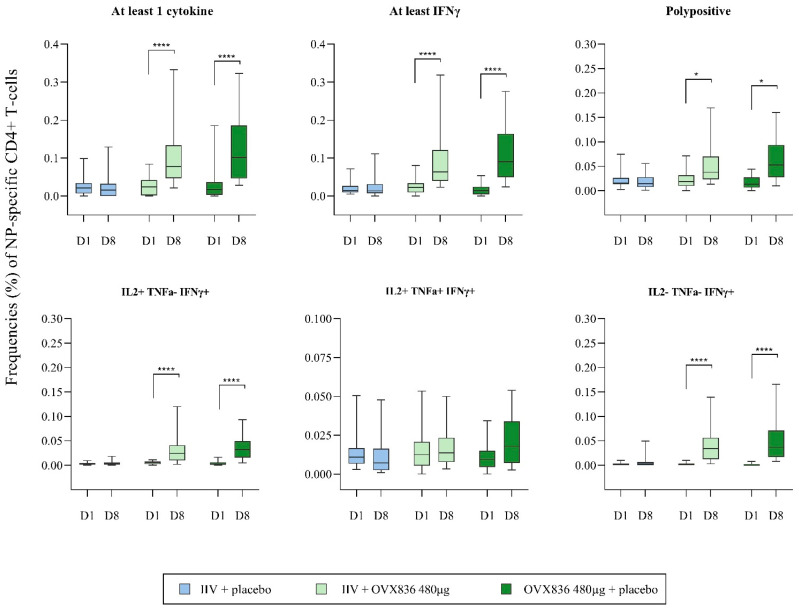
Box plot of the frequencies (%) of NP-specific CD4+ T-cells per total CD4+ T-cells positive for at least one of the three cytokines (among IFNγ, IL-2, and TNFα), for at least IFNγ (IFNγ only, but also IFNγ + IL-2, IFNγ + TNFα, and IFNγ + IL-2 + TNFα), for one, two (different combinations) or the three cytokines, and polypositive (at least two cytokines among IFNγ, IL-2, and TNFα) at baseline (Day 1) and Day 8 in the three treatment groups (per protocol cohort). Results are given as median (horizontal bar), interquartile interval (box), minimum, and maximum values. Intragroup comparisons (paired Student’s *t* tests; * *p* < 0.05; **** *p* < 0.0001).

**Figure 5 vaccines-13-00558-f005:**
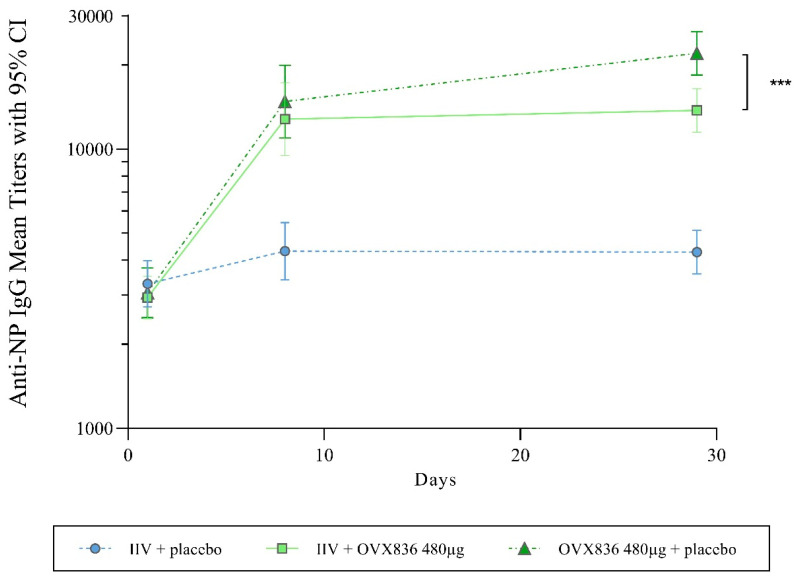
Anti-NP IgG titers at pre-injection baseline (Day 1), Day 8 (PBMC subset only), and Day 29 in the three treatment groups (per protocol cohort). Results are given as geometric mean titers and 95% confidence intervals. Intergroup comparison (ANOVA followed by Bonferroni’s tests; *** *p* = 0.0003).

**Table 1 vaccines-13-00558-t001:** Demographics and baseline characteristics of the study population, overall and by treatment group. Results are given as mean ± standard deviation for continuous variables and number (%) of participants for categorical variables.

	IIV + Placebo	IIV + OVX836 480 µg	OVX836 480 µg + Placebo	All Participants
	N = 60	N = 60	N = 60	N = 180
Age (year)	40.9 ± 12.9	36.6 ± 12.3	36.0 ± 12.9	37.8 ± 12.8
Weight (kg)	78.0 ± 14.6	76.0 ± 13.8	77.1 ± 13.5	77.0 ± 13.9
Height (cm)	172 ± 10	171 ± 8	173 ± 10	172 ± 9
BMI (kg/m^2^)	26.2 ± 3.7	26.0 ± 3.9	25.8 ± 3.8	26.0 ± 3.8
Sex (female)	34 (56.7)	36 (60.0)	37 (61.7)	107 (59.4)
Race (White)	49 (81.7)	49 (81.7)	48 (80.0)	146 (81.1)
Smoking status				
*Nonsmoker*	46 (76.7)	49 (81.7)	42 (70.0)	137 (76.1)
*Current smoker*	5 (8.3)	0 (0.0)	3 (5.0)	8 (4.4)
*Former smoker*	9 (15.0)	11 (18.3)	15 (25.0)	35 (19.4)

**Table 2 vaccines-13-00558-t002:** Solicited local and systemic symptoms (overall and severe) during the 7-day period post-administration and unsolicited adverse events (overall, severe, and relationship to the investigational product [IP]) during the 28-day period post-administration in the three treatment groups. Results are shown as the number (%) of subjects.

	Intensity	IIV + Placebo (N = 60)	IIV + OVX836 480 µg (N = 60)	OVX836 480 µg + Placebo (N = 60)
**Solicited local symptoms**				
Pain	All	31 (51.7)	52 (86.7)	48 (80.0)
	Severe	0 (0.0)	0 (0.0)	0 (0.0)
Redness	All	2 (3.3)	5 (8.3)	1 (1.7)
	Severe	0 (0.0)	0 (0.0)	0 (0.0)
Swelling	All	3 (5.0)	7 (11.7)	4 (6.7)
	Severe	0 (0.0)	0 (0.0)	0 (0.0)
**Solicited systemic symptoms**				
Arthralgia	All	16 (26.7)	15 (25.0)	15 (25.0)
	Severe	0 (0.0)	0 (0.0)	0 (0.0)
Fatigue	All	27 (45.0)	34 (56.7)	34 (56.7)
	Severe	1 (1.7)	0 (0.0)	0 (0.0)
Fever	All	3 (5.0)	2 (3.3)	1 (1.7)
	Severe	0 (0.0)	0 (0.0)	0 (0.0)
Headache	All	20 (33.3)	30 (50.0)	26 (43.3)
	Severe	0 (0.0)	0 (0.0)	1 (1.7)
Malaise	All	18 (30.0)	17 (28.3)	17 (28.3)
	Severe	0 (0.0)	0 (0.0)	0 (0.0)
Myalgia	All	27 (45.0)	39 (65.0)	33 (55.0)
	Severe	1 (1.7)	0 (0.0)	0 (0.0)
**Unsolicited adverse events**				
All		22 (36.7)	19 (31.7)	20 (33.3)
Severe		0 (0.0)	0 (0.0)	0 (0.0)
Related to the IP		7 (11.7)	13 (21.7)	8 (13.3)
Related to the IP and severe		0 (0.0)	0 (0.0)	0 (0.0)
**Serious adverse events**				
All		1 (1.7)	2 (3.3)	0 (0.0)
Related to the IP		0 (0.0)	0 (0.0)	0 (0.0)

**Table 3 vaccines-13-00558-t003:** Summary of the hemagglutination inhibition assay antibody response in the IIV + placebo and IIV + OVX836 480 µg groups at Day 29 (28 days after the vaccination) and achievement of the CPMP and FDA guidelines criteria (per protocol cohort).

Parameter	Strain	IIV + placebo (N = 48) n (%) [95% CI]	Criterion of the CPMP/FDA Guidelines Achieved	IIV + OVX836 480 µg (N = 49) n (%) [95% CI]	Criterion of the CPMP/FDA Guidelines Achieved
**HAI seroconversion rate**	A/Victoria/2570/2019 (H1N1) pdm09	38 (79.2) [65.0–89.5]	Yes/Yes	34 (69.4) [54.6–81.8]	Yes/Yes
	A/Darwin/9/2021 (H3N2)	17 (35.4) [22.2–50.5]	No/No	17 (34.7) [21.7–49.6]	No/No
	B/Austria/1359417/2021-like (B/Victoria lineage)	30 (62.5) [47.4–76.1]	Yes/Yes	32 (65.3) [50.4–78.3]	Yes/Yes
	B/Phuket/3073/2013-like (B/Yamagata lineage)	33 (68.8) [53.8–81.3]	Yes/Yes	36 (73.5) [58.9–85.1]	Yes/Yes
**HAI seroprotection** **rate**	A/Victoria/2570/2019 (H1N1) pdm09	45 (93.8) [82.8–98.7]	Yes/Yes	47 (95.9) [86.0–99.5]	Yes/Yes
	A/Darwin/9/2021 (H3N2)	39 (81.3) [67.4–91.1]	Yes/No	48 (98.0) [89.2–100.0]	Yes/Yes
	B/Austria/1359417/2021-like (B/Victoria lineage)	33 (68.8) [53.8–81.3]	No/No	34 (69.4) [54.6–81.8]	No/No
	B/Phuket/3073/2013-like (B/Yamagata lineage)	35 (72.9) [58.2–84.7]	Yes/No	43 (87.8) [75.2–95.4]	Yes/Yes
		**GMR** **(95% CI)**		**GMR** **(95% CI)**	
**HAI** **Day 29/Day 1 GMR**	A/Victoria/2570/2019 (H1N1) pdm09	7.02 (5.20–9.48)	Yes	6.79 (4.97–9.28)	Yes
	A/Darwin/9/2021 (H3N2)	2.80 (2.03–3.85)	Yes	3.17 (2.24–4.50)	Yes
	B/Austria/1359417/2021-like (B/Victoria lineage)	5.78 (4.29–7.77)	Yes	5.74 (4.60–7.17)	Yes
	B/Phuket/3073/2013-like (B/Yamagata lineage)	7.02 (5.19–9.49)	Yes	7.29 (5.74–9.26)	Yes

Results are given as N = total number of subjects, n (%) = number and percentage of subjects with the condition, Clopper–Pearson’s exact 95% confidence interval [95% CI], GMR = geometric mean ratio; and 95% CI = 95% confidence interval.

## Data Availability

The data presented in this study are available in this article (and Appendix A). The results have also been published on the ClinicalTrials.gov website.

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
