# Peer review of "Safety and Immunogenicity of OVX836, a Nucleoprotein-Based Universal Influenza Vaccine, Co-Administered with Fluarix® Tetra, a Seasonal Hemagglutinin-Based Vaccine"

_vaccines, 2025, doi:10.3390/vaccines13060558_

Round 1

Reviewer 1 Report

Comments and Suggestions for Authors

In this manuscript, Nicola and colleagues report results from a phase 2a clinical trial investigating co-administration of the nucleoprotein-based OVX836 and seasonal HA-based vaccines. The safety of OVX836 vaccine and its co-administration was demonstrated in a larger group size. The authors claim that no major interference occurred when these vaccines were co-administered.

The study is well-designed and comprehensively evaluated using inhibition assays, anti-NP IgG titers and IFN-gamma ELISPOT. The results demonstrate relatively high scientific soundness and the manuscript was well-written. The figure presentation in this manuscript, however, requires improvement. Some specific definitions and controls need to be described and discussed for clarification.

Comments:

  1. The overall quality of figures should be improved. Please maintain consistency in figure resolution, font, line weight, etc, in all figures.
  2. In Figure 2, I don’t find this bar plots with discontinuous y-axis show more and better information than traditional log-scale y-axis and scatter plots with geometric means. First, it is not clear for those data points in the upper panel, are they 280 or maybe 300? Second, I understand that the authors want to use the bar plots to show GM with 95% CI as well as GMR between groups. However, the lower bound CIs were not omitted in all plots. If they were included, the plots would also overcrowd with bars, CIs and scatter points and have worse visualization. I will suggest that just simply use the traditional scatter plots with log-scale y-axis to present titer data and indicate the GM with a thick line. I think the GMR and statistics are already mentioned in the tables within the main text or supplementary.
  3. For the result 3.3 discussing Table 3 and Figure 2, (first, please indicate in the main text where Table 3 is referenced) I think the authors didn’t include the description on defining seropositive, seronegative in this study. Do you use titer < 40 as seronegative? Additionally, I can tell from Figure 2, there are titers below  40 measured. I was wondering what the lower detection limit of this assay is in this study?
  4. Based on Table 2, all four influenza strains achieved primary endpoints because all GMRs exceed 2.5. For HAI seroprotection rates, IIV+placebo group failed to meet FDA criteria for A/Darwin, B/Austria and B/Phuket, while the IIV+OVX836 group met FDA criteria for A/Darwin and B/Phuket. I was wondering, are there adjuvant effects in OVX836?
  5. The novelty and significance in this study lie in the vaccine co-administration. The authors mentioned and discussed the absence or no major interference. In my opinion, to strengthen this conclusion, I recommend calculating, presenting in the Figure 2 or 3, and discussing the non-inferiority margins. Please see examples in doi: 10.1016/S0140-6736(21)02329-1.

Author Response

Reviewer 1

In this manuscript, Nicola and colleagues report results from a phase 2a clinical trial investigating co-administration of the nucleoprotein-based OVX836 and seasonal HA-based vaccines. The safety of OVX836 vaccine and its co-administration was demonstrated in a larger group size. The authors claim that no major interference occurred when these vaccines were co-administered.

The study is well-designed and comprehensively evaluated using inhibition assays, anti-NP IgG titers and IFN-gamma ELISPOT. The results demonstrate relatively high scientific soundness and the manuscript was well-written. The figure presentation in this manuscript, however, requires improvement. Some specific definitions and controls need to be described and discussed for clarification.

Thanks a lot to the reviewer for his/her global appreciation, relevant remarks, questions and suggestions.

Comments:

  1. The overall quality of figures should be improved. Please maintain consistency in figure resolution, font, line weight, etc, in all figures.

The overall quality of the figures has been improved and consistency has been ensured.

  1. In Figure 2, I don’t find this bar plots with discontinuous y-axis show more and better information than traditional log-scale y-axis and scatter plots with geometric means. First, it is not clear for those data points in the upper panel, are they 280 or maybe 300? Second, I understand that the authors want to use the bar plots to show GM with 95% CI as well as GMR between groups. However, the lower bound CIs were not omitted in all plots. If they were included, the plots would also overcrowd with bars, CIs and scatter points and have worse visualization. I will suggest that just simply use the traditional scatter plots with log-scale y-axis to present titer data and indicate the GM with a thick line. I think the GMR and statistics are already mentioned in the tables within the main text or supplementary.

Figure 2 has been changed in accordance with the reviewer’s suggestion: scatter plot with log-scale y-axis and GMT highlighted with a tick horizontal line.

  1. For the result 3.3 discussing Table 3 and Figure 2, (first, please indicate in the main text where Table 3 is referenced) I think the authors didn’t include the description on defining seropositive, seronegative in this study. Do you use titer < 40 as seronegative? Additionally, I can tell from Figure 2, there are titers below  40 measured. I was wondering what the lower detection limit of this assay is in this study?

Table 3 has now been referenced in the text. The hemagglutination inhibition assay (HAI) is described exhaustively in the Supplementary S2, which has been slightly updated to include the seronegative titre (<1:10 replaced by the number 5 in all GMTs and GMRs calculations), which corresponds to the limit of detection (LOD). Seropositivity is considered from titres ≥1:10. The definitions of seroconversion (negative pre-vaccination HAI titre and post-vaccination HAI titre ≥1:40, or fourfold increase in HAI titre between pre- and post-vaccination timepoints) and seroprotection (titres ≥1:40) are available in the M&M section.

  1. Based on Table 2, all four influenza strains achieved primary endpoints because all GMRs exceed 2.5. For HAI seroprotection rates, IIV+placebo group failed to meet FDA criteria for A/Darwin, B/Austria and B/Phuket, while the IIV+OVX836 group met FDA criteria for A/Darwin and B/Phuket. I was wondering, are there adjuvant effects in OVX836?

A limited adjuvant effect of OVX836 on the IIV cannot be ruled out in this study. GMTs at Day 29 were significantly higher in the IIV+OVX836 group compared to the IIV+placebo groups for the two A strains. However, there were no statistically significant differences between the IIV+OVX836 and the IIV+placebo groups in terms of seroconversion and seroprotection rates and GMRs. The study was powered to demonstrate the achievement of the CPMP criteria for influenza seasonal vaccines but not to show a difference between the two groups receiving the IIV (with or without OVX836). Moreover, no correction accounting for multiplicity testing has been implemented. P values just below 5% have therefore to be considered with caution as indicative only of potential statistically significant differences.

Just for information, another larger Phase 2a study (OVX836-006) has been performed to confirm the results of the current study with two different IIVs (Fluarix® Tetra and Afluria® Quad). OVX836 and IIV were administered separately in the two deltoids, but not in the same arm as in the current study. The results (soon to be published) have not shown any adjuvant effect of OVX836 on the IIVs but more importantly have confirmed the absence of any major interference.

  1. The novelty and significance in this study lie in the vaccine co-administration. The authors mentioned and discussed the absence or no major interference. In my opinion, to strengthen this conclusion, I recommend calculating, presenting in the Figure 2 or 3, and discussing the non-inferiority margins. Please see examples in doi: 10.1016/S0140-6736(21)02329-1.

This pilot Phase 2a study was primarily powered to demonstrate the achievement of the HAI CPMP criteria for influenza seasonal vaccines (minimum 50 subjects evaluable per group) but not to demonstrate the non-inferiority of IIV+OVX836 versus IIV+placebo. Therefore, we do not think it would be appropriate to introduce the notion of the non-inferiority margin in this manuscript.

Reviewer 2 Report

Comments and Suggestions for Authors

The stimulation of cellular immune responses in influenza vaccines, and in particular the incorporation of conserved proteins, is of vital importance to protect against a wide variety of existing and emerging new viruses. The OVX836 vaccine is a promising candidate for the development of a new vaccine. An exhaustive evaluation of this vaccine candidate, could help challenge and potentially shift the prevailing paradigm that protective immunity relies exclusively on neutralizing antibodies, highlighting the critical role of cellular and non-neutralizing immune mechanisms in vaccine-mediated protection. As well, I consider extremely important to evaluate this candidate combined with classical HA vaccines.

Overall, I consider the manuscript consistent and well presented. However, I would like to suggest a few minor changes and request clarification on some concerns.

  1. In the analysis of NP-specific IFN-γ spot-forming cells, how was the fold rise calculated? Was it determined as the mean of individual fold rises?
  2. In figure 3: Please, consider indicating significant differences within the same vaccinated-group between Day 1 and Day 8, for example, using asterisks or similar.
  3. In Figure 4: Please, consider labelling the y-axis as “frequencies (%) of NP-specific CD4+ T-cells”. Please, consider labelling each figure and its corresponding description in the legend for a clearer interpretation. Please, also consider indicating significant differences into the same vaccinated-group between the D1 and D8 using asterisks.
  4. Do you plan to assess the breadth of cellular immune responses at later time points?
  5. Could you please clarify the data presented in Figure 2? Specifically, “at least IFN-γ” refers to CD4⁺ T cells secreting only IFN-γ?
  6. It is well established that the protection conferred by the influenza nucleoprotein is primarily mediated by CD8⁺ cytotoxic T cell responses, which is a key objective in the development of NP-based influenza vaccines. However, in this study, the induction of such responses was not demonstrated. Considering your previous work, where you showed that your vaccine candidate at the 480 ug dose, is capable of eliciting CD8⁺ T cell responses, how do you justify the lack of significant differences observed in this case? Could this be attributed to technical aspects of the experimental setup, such as peptide/antigen stimulation conditions, or might it indicate a limitation in the ability of this particular combined formulation to induce CD8⁺ T cell responses? Have you considered assessing NP-specific IFN-γ CD8⁺ T cell responses using ELISPOT?
  7. Furthermore, do you consider that the absence of CD8⁺ T cell activation might compromise the rationale for the future use of this vaccine combination?
  8. It has been demonstrated that antibodies against influenza nucleoprotein can cross-react with the human hypocretin receptor 2, potentially leading to narcolepsy in individuals vaccinated with Pandemrix. This finding opened concerns about the practical implications of using an NP-based vaccine approach combined with classical HA-based vaccine development. Have you considered evaluating any narcolepsy-related signs, which have been associated with certain influenza vaccines and could potentially arise from the combination of both vaccines? Additionally, could you clarify how you plan to address potential safety issues related to this risk?

Author Response

Reviewer 2

The stimulation of cellular immune responses in influenza vaccines, and in particular the incorporation of conserved proteins, is of vital importance to protect against a wide variety of existing and emerging new viruses. The OVX836 vaccine is a promising candidate for the development of a new vaccine. An exhaustive evaluation of this vaccine candidate, could help challenge and potentially shift the prevailing paradigm that protective immunity relies exclusively on neutralizing antibodies, highlighting the critical role of cellular and non-neutralizing immune mechanisms in vaccine-mediated protection. As well, I consider extremely important to evaluate this candidate combined with classical HA vaccines.

Overall, I consider the manuscript consistent and well presented. However, I would like to suggest a few minor changes and request clarification on some concerns.

Thanks a lot to the reviewer for his/her appreciation, relevant remarks, questions and suggestions.

  1. In the analysis of NP-specific IFN-γ spot-forming cells, how was the fold rise calculated? Was it determined as the mean of individual fold rises?

The fold rise was actually determined as the mean of individual fold rises.

  1. In figure 3: Please, consider indicating significant differences within the same vaccinated-group between Day 1 and Day 8, for example, using asterisks or similar.

Statistically significant intragroup differences have now been highlighted in Figure 3 using symbols.

  1. In Figure 4: Please, consider labelling the y-axis as “frequencies (%) of NP-specific CD4+ T-cells”. Please, consider labelling each figure and its corresponding description in the legend for a clearer interpretation. Please, also consider indicating significant differences into the same vaccinated-group between the D1 and D8 using asterisks.

The y-axis of Figure 4 has been modified in accordance with reviewer’s suggestion and statistically significant intragroup differences have now been highlighted in Figure 3 using symbols.

  1. Do you plan to assess the breadth of cellular immune responses at later time points?

PBMCs were only sampled at Day 8 in this study, so that cellular response was only evaluated at this time point.

However, previous studies have shown that the cellular immune responses to OVX836  remain significant at Day 29 and are then decreasing between Day 29 and Day 180 (return to baseline values).

  1. Could you please clarify the data presented in Figure 4? Specifically, “at least IFN-γ” refers to CD4⁺ T cells secreting only IFN-γ?

‘At least IFNγ’ means CD4+ T-cells secreting IFNγ only but also IFNγ + IL2, IFNγ + TNFα and IFNγ + IL2 + TNFα.

The legend of Figure 4 has been clarified.

  1. It is well established that the protection conferred by the influenza nucleoprotein is primarily mediated by CD8⁺ cytotoxic T cell responses, which is a key objective in the development of NP-based influenza vaccines. However, in this study, the induction of such responses was not demonstrated. Considering your previous work, where you showed that your vaccine candidate at the 480 ug dose, is capable of eliciting CD8⁺ T cell responses, how do you justify the lack of significant differences observed in this case? Could this be attributed to technical aspects of the experimental setup, such as peptide/antigen stimulation conditions, or might it indicate a limitation in the ability of this particular combined formulation to induce CD8⁺ T cell responses? Have you considered assessing NP-specific IFN-γ CD8⁺ T cell responses using ELISPOT?

A statistically significant CD8+ response has been found in our previous study in young adults (Leroux-Roels et al., 2023) but not in elderly (Jacobs et al., 2024). The same methods and laboratories were used to evaluate the cellular response to OVX836 vaccination (with slight differences to process and freeze the PBMCs) in the current and the previous studies. Nevertheless, the differences between the current study and the one previously published are multifactorial:

  • Vaccinations were performed before the influenza season in the Leroux-Roels et al. (2023) study but during the influenza season in the present study. The circulation of natural influenza viruses during vaccination could have impacted the baseline CD8+ values.
  • Probably related to the vaccination period, we observed a high heterogeneity in the baseline CD8+ values in the present study with standard deviations being higher than the mean values themselves, resulting in coefficients of variation around 140%, while in the Leroux-Roels et al. (2023) study the standard deviations were just below the means, resulting in coefficients of variation around 90%. With a high variability at baseline it is more difficult to detect a significant CD8+ T-cell response.
  • There were more females (70.8%) enrolled in the Leroux-Roels et al. (2023) study than in the current study (59.4%). In our previous study (Jacobs et al., 2024) we have shown that females tended to show higher humoral and CMI responses to OVX836.
  • The study populations were enrolled in different hemispheres potentially leading to differences in historical influenza virus circulation: Northern hemisphere for the Belgian population in the Leroux-Roels et al. (2023) study versus Southern hemisphere for the Australian population in the present study.

In healthy adults, the CD4⁺/CD8⁺ ratio in peripheral blood is typically around 2 (Owen et al., 2013). Moreover, the amplitude of the CD8+ responses to vaccination are most of the time lower than that of CD4+ responses (Begue et al., 2022). This is also the case for the CMI responses to OVX836.

Regarding the detection of CD8+ T-cells in future clinical research we would like to consider other methods with increased sensitivity. This could be ELISPOT using 9-mer antigen stimulation, as you suggested, but also working on whole fresh blood instead of frozen PBMCs.

References

Begue S et al. (2022) Harmonization and qualification of intracellular cytokine staining to measure influenza-specific CD4+ T cell immunity within the FLUCOP consortium. Frontiers in Immunology, 10.3389/fimmu.2022.982887.

Jacobs B et al. (2024) Evaluation of Safety, Immunogenicity and Cross-Reactive Immunity of OVX836, a Nucleoprotein-Based Uni-versal Influenza Vaccine, in Older Adults. Vaccines 2024, 12, 1391, doi:10.3390/vaccines12121391.

Leroux-Roels I (2023) Immunogenicity, Safety, and Preliminary Efficacy Evaluation of OVX836, a Nucleoprotein-Based Universal Influenza A Vaccine Candidate: A Randomised, Double-Blind, Placebo-Controlled, Phase 2a Trial. Lancet Infect. Dis. 2023, 23, 1360–1369, doi:10.1016/S1473-3099(23)00351-1.

Owen et al. (2013). Kuby Immunology. New York: W. H. Freeman and Company. p. 40.

  1. Furthermore, do you consider that the absence of CD8⁺ T cell activation might compromise the rationale for the future use of this vaccine combination?

CD8+ T-cell activation remains an important feature for the OVX836 vaccine either considered as a standalone vaccine or in combination with IIVs. It should be emphasized that the absence of CD8+ following concomitant administration of OVX836+IIV was not attributable to the co-administration since the same observation was also made in the OVX836+placebo group.

No correlate of protection has yet been demonstrated for OVX836.  Therefore it is unclear to date which surrogate markers could be used to determine the efficacy of OVX836.  For this reason Osivax is planning a large proof-of-concept vaccine efficacy Phase 2b trial to demonstrate the clinical benefit of OVX836 as measured through absolute vaccine efficacy of OVX836 in healthy adults (during the Winter 2025/2026 in the North Hemisphere).

Apart from the desirable CD8+ T-cell response, both CD4+ T-cell and anti-NP IgG could also play an important role in the protection provided by the vaccine:

Importance of the CD4+ T-cell response:

  • A human challenge study has demonstrated that pre-existing CD4+ T-cells responding to internal influenza proteins were associated with lower virus shedding and less severe illness (Wilkinson et al., 2012; Sridhar et al., 2013).
  • Around 1980, 63 volunteers were challenged with the re-emerging Influenza H1N1 A/USSR/77 virus. In those without pre-existing antibody responses but with detectable T-cell responses, there was a reduction in nasal virus shedding but not in symptoms (McMichael et al., 1983). The T-cell responses were probably mostly mediated by CD8+ T-cells. A study performed thirty years later drew similar conclusions, but by then it was possible to distinguish CD4+ and CD8+-specific T-cell responses and the former were as, or more, protective than the CD8+ T-cell responses (Wilkinson et al., 2012). In addition, influenza-like-illness was less severe in volunteers with pre-existing T-cell responses.
  • In a third large community study in Hong Kong during the 2009 pandemic and subsequent seasons, protection from infection was observed in participants with both, pre-existing CD4+ and CD8+ T-cell responses, but effects on severity were not measured (Tsang et al., 2022).

Importance of the anti-NP IgG response: Some authors have suggested that anti-NP IgG were not neutralizing by themselves, but could bind to virus-infected cells and mediate protection through Fc-dependent functions (Rak et al., 2023; Carragher et al., 2008; Jegaskanda et al., 2017; Jegaskanda et al., 2013).

References

Carragher DM et al. (2013) A Novel Role for Non-Neutralizing Antibodies against Nucleoprotein in Facilitating Resistance to Influenza Virus. J. Immunol. Baltim. Md 1950, 181, 4168–4176, doi:10.4049/jimmunol.181.6.4168.

Jegaskanda S et al. (2017) Fc or Not Fc; That Is the Question: Antibody Fc-Receptor Interactions Are Key to Universal Influenza Vaccine Design. Hum.Vaccin.Immunother., 13, 1–9, doi:10.1080/21645515.2017.1290018.

Jegaskanda S et al. (2013) Standard Trivalent Influenza Virus Protein Vaccination Does Not Prime Antibody-Dependent Cellular Cytotoxicity in Macaques. J. Virol., 87, 13706–13718, doi:10.1128/JVI.01666-13.

McMichael AJ et al. (1983) Cytotoxic T-cell immunity to influenza. N Engl J Med, 309, 13-17, doi:10.1056/NEJM198307073090103.

Rak A et al. (2023) Nucleoprotein as a Promising Antigen for Broadly Protective Influenza Vaccines. Vaccines 2023, 11, 1747, doi:10.3390/vaccines11121747.

Sridhar S et al. (2013) Cellular immune correlates of protection against symptomatic pandemic influenza. Nat Med 19, 1305-1312, doi:10.1038/nm.3350.

Tsang TK et al. (2022) Investigation of CD4 and CD8 T cell-mediated protection against influenza A virus in a cohort study. BMC Med, 20, 230, doi:10.1186/s12916-022-02429-7.

Wilkinson TM et al. (2012) Preexisting influenza-specific CD4+ T cells correlate with disease protection against influenza challenge in humans. Nat Med, 18, 274-280, doi:10.1038/nm.2612.

  1. It has been demonstrated that antibodies against influenza nucleoprotein can cross-react with the human hypocretin receptor 2, potentially leading to narcolepsy in individuals vaccinated with Pandemrix. This finding opened concerns about the practical implications of using an NP-based vaccine approach combined with classical HA-based vaccine development. Have you considered evaluating any narcolepsy-related signs, which have been associated with certain influenza vaccines and could potentially arise from the combination of both vaccines? Additionally, could you clarify how you plan to address potential safety issues related to this risk?

Osivax is aware and sensitive to this very important safety question. However, at this point of the clinical development program it is probably a wide premature to evaluate a potential rare safety risk like narcolepsy in a limited clinical safety database. However, Osivax will continue to implement routine oversight by collecting safety data including AESIs to capture any potential safety signal during the conduct of its clinical trials with OVX836.  

In the past an increase in reported cases of narcolepsy Type 1 (NT1) has been associated with the adjuvanted A(H1N1)-pdm09 vaccine (Pandemrix®) administered during the global A(H1N1) influenza pandemic in 2009, but not with the MF59-adjuvanted A(H1N1)pdm09 vaccine (Focetria®). Among factor(s) and mechanism(s) potentially involved, anti-NP and/or anti-HA responses were proposed to contribute through molecular mimicry, and the role of adjuvant has also been investigated (Ahmed et al., 2015; Buonocore et al., 2022; Ayoub et al., 2024; Xu et al., 2024). This theory has been questioned and/or continues to be further evaluated. It is hypothesized that a potential bias could have affected the conclusions of the initial epidemiological analyses. In this regard, more recent studies did not confirm the vaccine-associated risk incidence initially established across several European countries. Overall, the different investigations, while supporting an auto-immune aetiology for NT1, do not support a direct role for influenza NP/HA-induced responses. Moreover, the strength of the causal relationship between Pandemrix® and narcolepsy is under discussion. As influenza-linked inflammation caused by the natural infection could also contribute to the narcolepsy onset, the risk present at the time of mass vaccination campaigns could have resulted from the co-circulation of the wild type H1N1 at the time of vaccination, or shortly before it. In conclusion, the risk of vaccine-associated narcolepsy is still to be firmly established.

Ahmed SS et al. (2015) Antibodies to influenza nucleoprotein cross-react with human hypocretin receptor 2. www.ScienceTranslationalMedicine.org 1 July 2015 Vol 7 Issue 294 294ra105.

Ayoub I et al. (2024) Infection, vaccination and narcolepsy type 1: Evidence and potential molecular mechanisms. Journal of Neuroimmunology. Volume 393, 578383.

Buonocore SM et al. (2022) Narcolepsy and H1N1 influenza immunology a decade later: What have we learned? Frontiers in Immunology, 10.3389/fimmu.2022.902840.

Xu W et al. (2024) The Role of T Cells in the Pathogenesis of Narcolepsy Type 1: A Narrative Review. International Journal of Molecular Sciences. 25, 11914. https://doi.org/10.3390/ijms252211914.

Reviewer 3 Report

Comments and Suggestions for Authors

We know, that effective universal influenza vaccine will be more progressive and exclude necessity of annual vaccination. This manuscript  was devoted of this aim. It were investigated not only humoral immunity after vaccination, but also cell - mediated immunity. The phase 2a of clinical trails was organized according all rulls.

However the manuscript have to improve in the sections "Metods" and "Results".

In section "Methods" not indicate the base strains for nucleoprotein vaccine OVX836. Need to clarification.

In section "Results":

1) On the page 7 need to correction of influenza strain names: A/Darwin/9/2021 (H3N2), B/Austria/1359417/2021, B/Phuket/3073/2013 and A/Victoria/2570/2019 (H1N1)pdm09. 

Need to write full strain names instead short.

2) Appropriate correction need to do in the legend of Figure 3 (page 8 line 6).

3)  Table 3. Need to add (B/Victoria linage) for strain B/Austria/1359417/2021 in 3 boxes.

Author Response

Reviewer 3

We know, that effective universal influenza vaccine will be more progressive and exclude necessity of annual vaccination. This manuscript  was devoted of this aim. It were investigated not only humoral immunity after vaccination, but also cell - mediated immunity. The phase 2a of clinical trials was organized according all rules.

Thanks a lot to the reviewer for his/her global appreciation, relevant remarks, questions and suggestions.

However the manuscript have to improve in the sections "Methods" and "Results".

In section "Methods" not indicate the base strains for nucleoprotein vaccine OVX836. Need to clarification.

OVX836 is a recombinant protein that includes the full-length nucleoprotein (NP) of the A/WSN/1933(H1N1) influenza virus. This information is already mentioned in the introduction and has been recalled in the M&M section.

In section "Results":

1) On the page 7 need to correction of influenza strain names: A/Darwin/9/2021 (H3N2), B/Austria/1359417/2021, B/Phuket/3073/2013 and A/Victoria/2570/2019 (H1N1)pdm09. 

Need to write full strain names instead short.

The full names of the influenza A and B strains have been provided in the ‘Results’ section.

2) Appropriate correction need to do in the legend of Figure 2 (page 8 line 6).

The full names of the influenza A and B strains have also been provided in the legend of Figure 2.

3)  Table 3. Need to add (B/Victoria linage) for strain B/Austria/1359417/2021 in 3 boxes.

The full names of the influenza A and B strains have also been provided in Table 3.
